# Unraveling Spatial and Temporal Heterogeneities of Very Slow Rock-Slope Deformations with Targeted DInSAR Analyses

**Chiara Crippa [1],\*** , **Federico Franzosi [1]** , **Mattia Zonca [1]** , **Andrea Manconi [2]** ,
**Giovanni B. Crosta [1]** , **Luca Dei Cas [3]** and **Federico Agliardi [1]**

1   Department of Earth and Environmental Sciences, University of Milano-Bicocca, 20126 Milano, Italy;
    f.franzosi@campus.unimib.it (F.F.); m.zonca7@campus.unimib.it (M.Z.);
    giovannibattista.crosta@unimib.it (G.B.C.); federico.agliardi@unimib.it (F.A.)
2   Department of Earth Sciences, ETH Zurich, 8092 Zurich, Switzerland; andrea.manconi@erdw.ethz.ch
3   Centro di Monitoraggio Geologico, ARPA Lombardia, 23100 Sondrio, Italy; l.deicas@arpalombardia.it
\*   Correspondence: c.crippa16@campus.unimib.it

**Abstract:** Spaceborne radar interferometry is a powerful tool to characterize landslides at local and regional scales. However, its application to very slow rock slope deformations in alpine environments (displacement rates < 5 cm/year) remains challenging, mainly due to low signal to noise ratio, atmospheric disturbances, snow cover effects, and complexities resulting from heterogeneous displacement in space and time. Here we combine SqueeSAR^TM data, targeted multi-temporal baseline DInSAR, GPS data, and detailed field morpho-structural mapping, to unravel the kinematics, internal segmentation, and style of activity of the Mt. Mater deep-seated gravitational slope deformation (DSGSD) in Valle Spluga (Italy). We retrieve slope kinematics by performing 2D decomposition (2D InSAR) of SqueeSAR^TM products derived from Sentinel-1 data acquired in ascending and descending orbits. To achieve a spatially-distributed characterization of DSGSD displacement patterns and activity, we process Sentinel-1 A/B images (2016-2019) with increasing temporal baselines (ranging from 24-days to 1-year) and generate several multi-temporal interferograms. Unwrapped displacement maps are validated using ground-based GPS data. Interferograms derived with different temporal baselines reveal a strong kinematic and morpho-structural heterogeneity and outline nested rockslides and active sectors, that arise from the background displacement signal of the main DSGSD. Seasonal interferograms, supported by GPS displacement measurements, reveal non-linear displacement trends suggesting a complex response of different slope sectors to rainfall and snowmelt. Our analyses clearly outline a composite slope instability with different nested sectors possibly undergoing different evolutionary trends towards failure. The results herein outline the potential of a targeted use of DInSAR for the detailed investigation of very slow rock slope deformations in different geological and geomorphological settings.

**Keywords:** deep-seated gravitational slope deformation; slow rock-slope deformation; kinematics; landslide activity; heterogeneity; DInSAR processing; SqueeSAR; GPS

## 1. Introduction

Giant, very slow rock slope deformations known as deep-seated gravitational slope deformations (DSGSD) are widespread in high mountain environments worldwide [1–5]. The evolution of these phenomena is strongly constrained by inherited geological structure and long-term climatic forcing and occurs through progressive damage accumulation. This can lead to progressive strain localization

and nucleation of secondary landslides, nested within the main DSGSD mass, that become faster and more sensitive to hydrological processes as damage accumulates [6–8].

DSGSDs creep slowly over thousands of years, affecting entire valley slopes and displacing rock volumes up to billions of cubic meters [4,9,10]. Therefore, even if displacement rates are low (usually smaller than 5 cm/year [11–13]) and apparently steady, DSGSDs result in large cumulative deformations that are mirrored by different morpho-structural field evidence. These include extensional features as double-crested ridges, trenches, scarps, counterscarps, and half-grabens, usually dominating in the upper slope sectors, and compressional features like bulging, thrusting and folding, and nested large landslides in the middle-lower sectors. The kinematic interpretation of individual features and their associations provide critical information on the overall slope deformation mechanism and the patterns of both distributed and localized strain, reflecting deep-seated deformation mechanisms [14].

Slow rock slope deformations cause severe degradation of rock mass strength, damage to infrastructures [15–17] and possible evolution of slope sectors towards faster rates [6]. A reliable characterization of DSGSD mechanisms and activity is thus crucial to identify possible failure scenarios and for risk reduction. Nevertheless, DSGSDs are usually characterized by: (a) strongly heterogeneous displacement patterns, associated with complex mechanisms (strain partitioning, damage localization and/or secondary landslides nested at different depths); (b) variable and often unknown trends of activity and sensitivity to external forcing [4,6,8]. In order to investigate these points, displacements must be characterized in a spatially-distributed fashion and with a sufficiently high rate of temporal sampling.

Among the available in situ and remote landslide mapping and monitoring techniques, spaceborne differential radar interferometry (DInSAR) allows characterizing ground deformation rates from a few millimeters to centimeters per year [18], maximizing the spatial and temporal coverage at a relatively low cost [12,19]. Persistent-scatterer interferometry (PSI) techniques allow extracting coherent information on line-of-sight (LOS) displacement rates at point targets, even when spatial baselines and revisit times of satellite platforms were not favorable [20]. Thus, PSI became a standard tool for landslide investigation, also applied to a regional appraisal of DSGSDs [12,13]. However, the application of point-like data extracted by PSI processing chains to the detailed analysis of slow-moving DSGSDs in high mountain environments remains challenging. This is mainly due to low signal to noise ratio, severe atmospheric disturbances and snow cover effects that limit the occurrence of scatterers with sufficiently high coherence. This results in a sparse point cloud that is not dense enough to fully analyze the spatial displacement pattern.

Such specific problems can be overcome only by targeting DInSAR on the specific features and typical spatial-temporal scales of very slow rock slope deformations, integrating: (a) DInSAR processing over multiple temporal baselines; (b) detailed field morpho-structural constraints; (c) ground-based measurements.

In this perspective, we investigate the Mt. Mater DSGSD in Valle Spluga (Italian Central Alps), which affects a 1300 m-high glacial valley flank over an area of 3 km$^2$. The slope impends over Madesimo village and ski resort and, according to published data [12,13], it is one of the fastest DSGSD in the Lombardia region. In order to unravel the kinematics, segmentation, and style of activity of Mt. Mater DSGSD, we integrate available SqueeSAR$^{TM}$ products, site-specific DInSAR processing of Sentinel-1 radar images, GPS data, and detailed morpho-structural information from aerial photointerpretation and field surveys. Our results outline a strong spatial segmentation and suggest a complex seasonal pattern of slope deformations at Mt. Mater, providing important guidance to further modeling activities aimed at risk reduction.

## 2. Case Study: Mt. Mater DSGSD

### 2.1. Geology and Geomorphology

The slope is located on the eastern flank of the upper Valle Spluga (Lombardia, Central Italian Alps) and impends over Madesimo village and tourist resort (Figure 1).

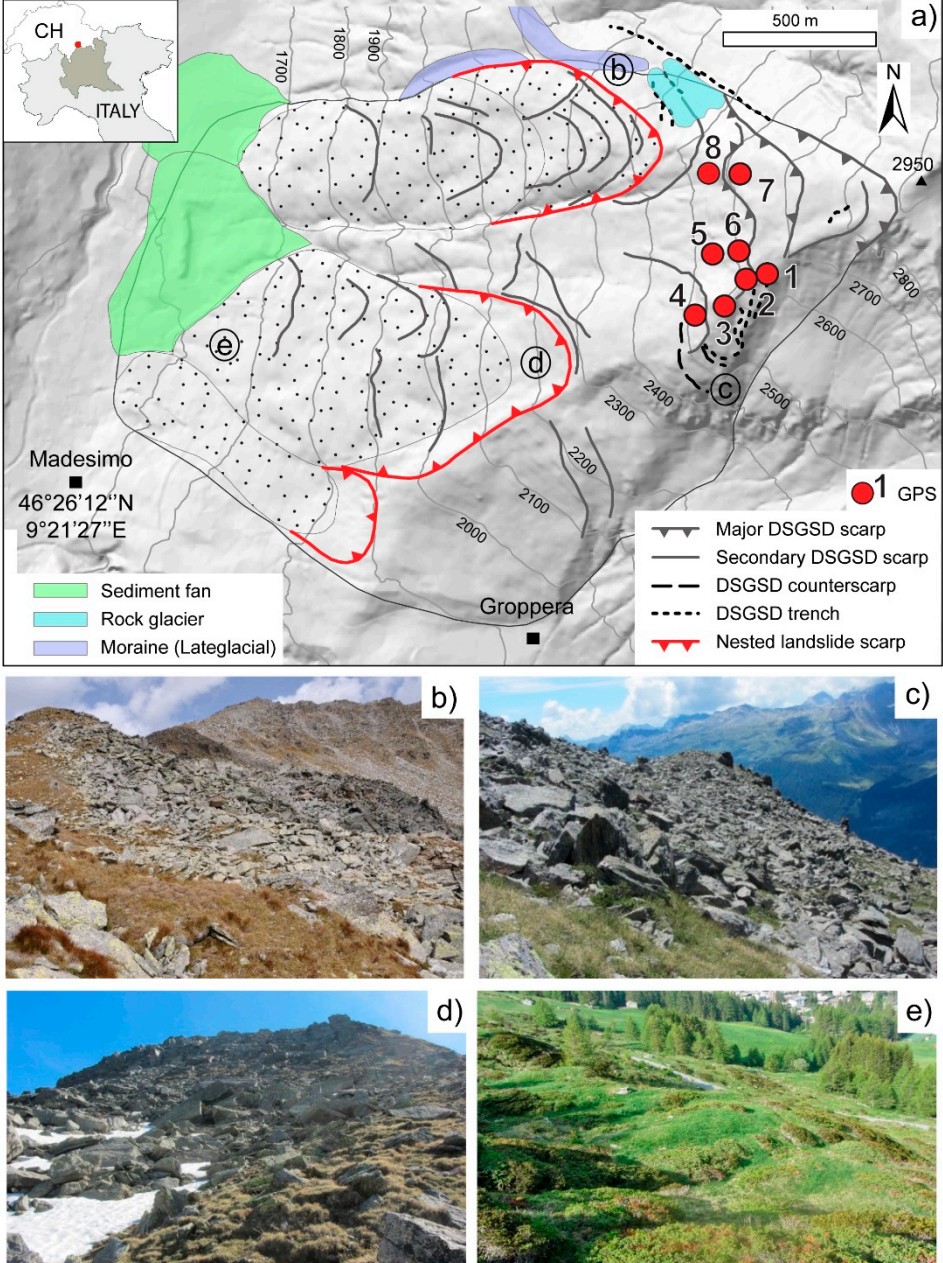

**Figure 1.** Main features of the Mt. Mater deep-seated gravitational slope deformation (DSGSD). (**a**) Simplified map portraying the main morpho-structural features associated with DSGSD and nested large landslides. GPS benchmark locations are outlined. (**b**) tranches on the S side of the main DSGSD body; (**c**) main scarp of the northern nested landslide (cross-section in Figure 6); (**d**) damaged rock mass at the top of the southern nested landslide; (**e**) blocky accumulation at the southern nested landslide toe.

The high relief slope (1550 to 3000 m a.s.l.) ranges in inclination between 33° (<2500 m a.s.l.) and 25° (>2500 m a.s.l.). The slope is made of metamorphic rocks of the Suretta and Tambò Penninic nappes [21], with dominant mica schist and paragneiss of the Stella-Timun complex (Suretta nappe) and limited outcrop of Mesozoic metasedimentary rocks at the slope toe, marking the contact with the underlying Tambò nappe. Rocks underwent a polyphase alpine tectono-metamorphic evolution with four main deformation stages since the Paleocene [22,23]. The first two stages involved the regional nappe structure and resulted in tight isoclinal folds associated with a pervasive foliation moderately dipping to the East. Locally, the third stage refolded previous structures with a subvertical N-verging

axial plane [23,24]. Later ductile-brittle deformation resulted in the development of NNW-SSE trending normal faults cutting nappe boundaries [22] and in N-S and E-W trending master fractures. Tectonic strain and fabric are strongly heterogeneous due to the widespread occurrence of ductile shear zones, affecting the variability of rock strength and deformability in the brittle field.

The geomorphology of the upper Valle Spluga was strongly imprinted by Quaternary glaciations, especially by the Last Glacial Maximum (LGM [25]), that carved steep slopes in basement rocks with local reliefs up to 1500 m. After post-LGM deglaciation, landscape development was controlled by Lateglacial glacier re-advances [26], testified by well-preserved glacial cirques, steps and moraine ridges below 2350 m a.s.l. (Figure 1). During and following Lateglacial, extensive periglacial processes strongly affected rock degradation, slope stability, and sediment redistribution, as testified by widespread scree deposits and intact rock glaciers above 2400 m a.s.l. [27].

*2.2. Gravitational Morpho-Structures*

The post-glacial destabilization of the entire slope by means of deep-seated gravitational slope deformation (DSGSD) is testified by evident morpho-structural features [28,29], that affect the slope from the toe to the crest over an area exceeding 3 km$^2$. These features have been recognized, mapped and interpreted through a geological and geomorphological investigation of the area, integrating 1:5000 field surveys, aerial photo-interpretation (stereo-photo coverage; TEM 1981-83; ortho-photos: 2000, 2003, 2007, 2012, 2015) and high-resolution digital elevation models (5 m resolution).

Above 2500 m a.s.l., these features include dominant scarps, suggesting limited or negligible rotational movements within the DSGSD mass. At 2900 m a.s.l., the slope is cut by a sharp triangular head scarp with a vertical downthrow of about 40 m (Figure 1a). Laterally, the scarp trace can be followed continuously to the N and S, suggesting that the DSGSD is bounded by a relatively well-developed basal shear zone. Between 2900 and 2500 m a.s.l., the slope is cut by several steep persistent N-S trending scarps, that extend to the south crossing the ridge with the Val Groppera. Here, NE-SE trenches and NW-SE counterscarps with a maximum length of few tens of meters define a small gravitational graben (Figure 1b).

Moving downslope, shallower arcuate scarps mark the transition to two nested large landslides, affecting the slope between 2400 m a.s.l. to the toe (Figure 1). In these sectors, rock masses are progressively more damaged moving downslope and become almost entirely crushed at 1900–2000 m a.s.l. Nested landslides are bounded by curve-shaped, highly fractured main scarps with vertical downthrow up to 20–25 m (Figure 1c,d). Several secondary scarps occur between 2200 and 2450 m a.s.l. within the landslide masses, suggest their internal segmentation in sectors with different degrees of damage (Figure 1a). Both the nested landslides have highly displaced toes, pushed out towards the valley floor through a series of slope-parallel lobes associated with debris sliding and a localized collapse of large landslide toes (Figure 1e). The main scarp of the northern nested landslide partially dismantled a moraine ridge at about 2300 m a.s.l., while DSGSD morpho-structures interact with intact rock-glaciers. This suggests that large-scale rock-slope instability was active in different stages after the Lateglacial.

## 3. Materials and Methods

We characterized the DSGSD using products of different spaceborne differential SAR interferometry (DInSAR) techniques. To analyze different aspects of the phenomenon, we used both point-like information from SqueeSAR$^{TM}$ data (2015–2017) and spatially distributed information from specially processed multi-temporal baseline interferograms. We integrated this information with GPS ground measurements covering a period of 4 years from 2014 to 2019 (Figure 2).

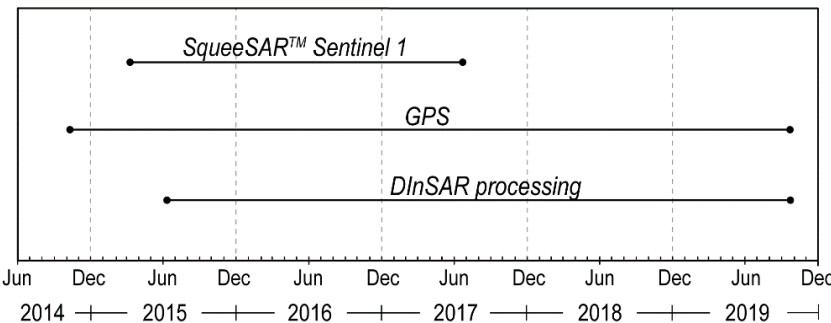

**Figure 2.** Time windows and temporal baselines covered by remote sensing and monitoring data.

## 3.1. SqueeSAR^TM Data

In order to test the potential of commercially available PSI data processed over large areas for the detailed characterization of a very slow rock slope deformation, we used SqueeSAR^TM data (TRE Altamira; Figure 3a,b). These were derived from Sentinel 1A/B SLC radar images (C-band, wavelength: 5.56 cm), acquired in TOPS Interferometric Wide swath (IW) mode [30] between March 2015 and July 2017 (revisit: 12 days, 6 days since March 2017; Supplementary Tables S1 and S2) in both ascending (track = 15; orbit azimuth $\delta$ = 10.23°; mean LOS angle $\theta$ = 41.99°) and descending geometries (track = 168; $\delta$ = 8.99°; $\theta$ = 41.78°). SqueeSAR^TM data report LOS displacement rates ($V_{LOS}$). When the true displacement rate vector deviates from the LOS, InSAR sensitivity decreases making the interpretation of slow movements challenging. The Mt. Mater slope, facing to the W (mean aspect: 262°) with a mean slope gradient of 28°, is favorably oriented to the satellite LOS of descending orbits, able to catch slope movements with the highest sensitivity and without significant geometrical distortions.

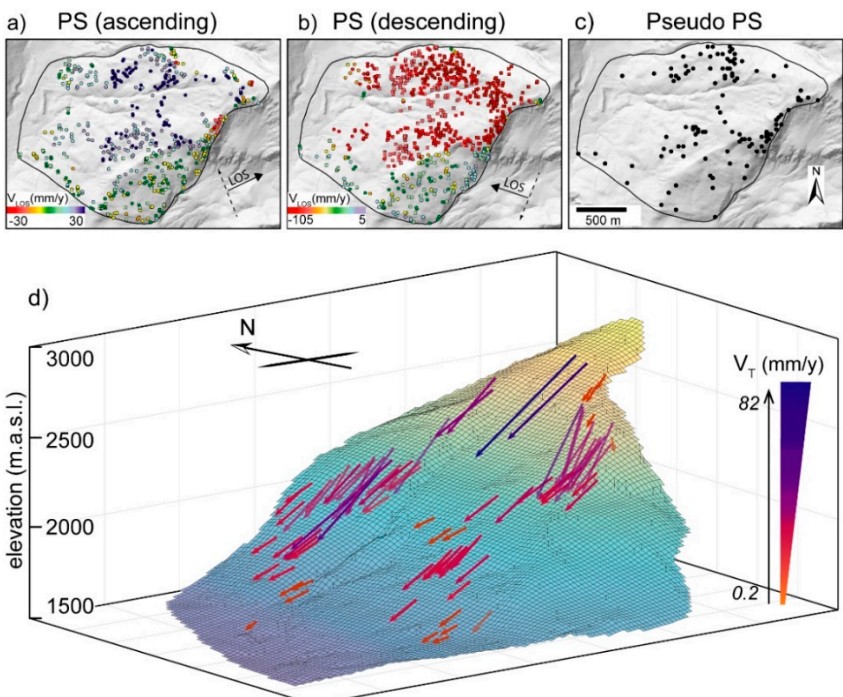

**Figure 3.** Kinematics of the Mt. Mater DSGSD derived from SqueeSAR^TM data analysis. (**a**,**b**) permanent (PS) and distributed scatterers (DS), classified by line-of-sight (LOS) displacement rate ($V_{LOS}$); (**c**) pseudo-PS derived by combining data from ascending and descending orbit; (**d**) 3D visualization of the 2D total displacement vector $T$, derived by the 2D decomposition of LOS velocities at pseudo-PS locations. Vector color and length are classified by $T$ vector velocity ($V_T$).

To maximize the exploitation of SqueeSAR$^{TM}$ data (Figure 3a,b), we integrated the information provided by ascending and descending acquisitions by decomposing the 2D velocity vector in the E-W trending vertical plane (2DInSAR; [14,31,32]). We developed a Matlab$^{TM}$ script to discretize the study area into a regular square grid (cell size: 25 m) and, for each cell, we averaged the LOS velocity values of PS/DS belonging to the same acquisition geometry. The average LOS velocity value computed in each cell for the two geometries was then assigned to pseudo-PS points, corresponding to the cell centroids (Figure 3c). For each pseudo-PS with both ascending and descending mean LOS velocity values, we extracted the E-W horizontal (*Ve*, Equation (1)) and vertical (*Vz*, Equation (2)) displacement rate components, as well as the magnitude (*V$_T$*, Equation (3)) and inclination (*dipT*, Equation (4)) of the 2D displacement rate vector *T*.

$$V_e = \frac{V_d \cos \theta_a - V_a \cos \theta_d}{\sin(\theta_a + \theta_d)} \tag{1}$$

$$V_z = \frac{V_d \sin \theta_a - V_a \sin \theta_d}{\sin(\theta_a + \theta_d)} \tag{2}$$

$$V_T = \sqrt{V_z{}^2 + V_e{}^2} \tag{3}$$

$$dipT = \cos^{-1} \frac{V_e}{V_T} \tag{4}$$

where: $V_a$ and $V_d$ are the ascending and descending LOS velocities (mm/year), and $\theta_a$ and $\theta_d$ are the mean LOS angles for Sentinel 1 platform in the two acquisitions geometries.

### 3.2. DInSAR Processing

To further investigate the spatial and temporal patterns and heterogeneities of DSGSD displacements, we used the software SNAP (ESA Sentinel Application Platform 7.0.0 [33]) to generate 61 interferograms with increasing temporal baselines, ranging from 24 days to 1 year in the period between June 2016 and October 2019 (Figure 2 and Supplementary Table S3). Shorter baseline interferograms (6, 12 days) have been discarded because they are only able to capture fast displacements of scree deposits or periglacial features. To reduce speckle noise and better outline phase signatures in the interferograms, we applied multi-looking factors of 1 (azimuth) and 4 (range) and phase filtering techniques (Goldstein phase filtering [34]). For those interferograms with good signal to noise ratio and high coherence, we obtained phase-unwrapped displacement maps using the minimum cost flow algorithm (MCF, [35]) implemented in the SNAPHU software plugin [36].

Short temporal baseline (24-day) interferograms are frequently affected by atmospheric disturbances in the early June and September-October periods, due to the association of sharp topography and daily temperature variations. Thus, they were carefully selected to avoid submitting misleading information to the unwrapping procedure.

Seasonal interferograms with temporal baselines of several months (Figure 2) were processed considering snow cover occurrence from nivometric data (Campodolcino Meteo Station) and Landsat 8 (OLI) C1 images. Suitable pairs of snow-free images were selected to generate interferograms representative of displacements accumulated from June to October (hereafter summer interferograms) and from October to June (hereafter snow cover period interferograms).

Finally, annual interferograms were targeted to outline persistent (pluri-annual) displacement signals. We considered pairs of images with temporal baseline spanning 1-year (±6 days), rolling over the period June 2016–October 2019.

### 3.3. GPS Data

We considered 8 GPS benchmarks (operated by ARPA Lombardia), installed in the upper sector of the slope above 2500 m.a.s.l. (Figure 1). Periodical differential GNSS measurements (about 3 surveys per year), carried out using double-frequency GNSS Leica GS09 receivers, started in October 2014 for station 3 and in June 2015 for the other ones. Considering instrumentation specifications and the distance to

the master control station (~1.5 Km), measurements are affected by a static (post-processed) horizontal standard error of 3–5 mm + 0.5 ppm and a vertical one of 6–10 mm + 1 ppm. No further corrections have been applied for environmental sources of error affecting measurements in field conditions.

## 4. Data and Results

We characterized the global kinematics (i.e., sliding mechanism) of the Mt. Mater DSGSD by exploiting the available SqueeSAR$^{TM}$ dataset based on Sentinel-1 radar images (Figure 3a,b), from which we derived the products of 2D displacement rate vector decomposition (2DInSAR). The spatial distribution of the *dipT* variable along the slope allows identifying sectors undergoing lowering, slope-parallel translation or bulging/uplift, thus providing a consistent figure of global slope kinematics [31].

For Mt. Mater, our results indicate an average *dipT* value of about 30° over the slope, while higher values (around 80°) can be detected only close to the steep DSGSD upper scarps and to the headscarps of nested large landslides. These values and their spatial distribution provide robust insights into the global kinematics of the DSGSD. This is characterized by dominant translational sliding, with rotational sliding components associated with the head sectors of the nested large landslides mapped in the field (Figures 1 and 3d).

$V_T$ values (Figure 3d) are generally higher in the upper DSGSD sectors, reaching 80 mm/year above the curved scarp at around 2600 m a.s.l. (Figure 1a) and testifying the ongoing active deformation of the entire slope. At and above the head sectors of the two nested large landslides, $V_T$ values range between 30 and 45 mm/year (head of northern landslide; Figure 1a,b) and 25–30 mm/year (above the head of southern landslide; Figure 1a,d). Displacement rates fade to 1–5 mm/year moving towards the slope toe, where landslide materials become more crushed and crumpled and/or the true displacement vector may significantly deviate from the radar line-of-sight.

However, the sparse nature of these data hampers a sound interpretation of the spatial pattern of measured displacements with respect to the scale (i.e., shallow vs deep-seated) and heterogeneity of slope failure mechanisms characteristic of different slope sectors. Thus, to better investigate displacement patterns along the slope, we analyzed the multi-temporal interferograms generated through targeted DInSAR processing across the period June 2016-October 2019 (Figure 4).

In all the processed interferograms, a low coherence area (<0.4) due to patchy vegetation and thermal stratification of the valley air, hampers the possibility to detect deformations close to the slope toe. This effect becomes more severe when increasing the temporal baseline from 24 days to 1-year, in which, the low coherence sector extends up to the middle slope sector because of increasing temporal decorrelation.

Interferograms with a baseline shorter than 24 days (i.e., 6, 12 days) do not show significant displacements, except for localized superficial debris covers or periglacial forms. From 24 days on, a triangular-shaped moving sector emerges from an almost homogeneous ground displacement signal (Figure 4a). It extends between the N-S curvilinear scarp trending at 2500 m a.s.l. and the DSGSD headscarp at 2900 m a.s.l. (Figure 1) and is made of damaged rock with extensive debris cover and periglacial features. This is the fastest slope sector, as it always decorrelates in longer baseline interferograms and is characterized by average displacement rates of 10 mm/month (Figure 4b).

These values are consistent with data provided by GPS benchmarks, all located within the upper slope sector between 2500 and 2650 m a.s.l. (Figures 1 and 5a). As few GPS measurements were carried out during each year, for this comparison we referred to displacement rates averaged over the entire time span covered by GPS (2015 to 2019), that were projected along the descending Sentinel-1 satellite LOS.

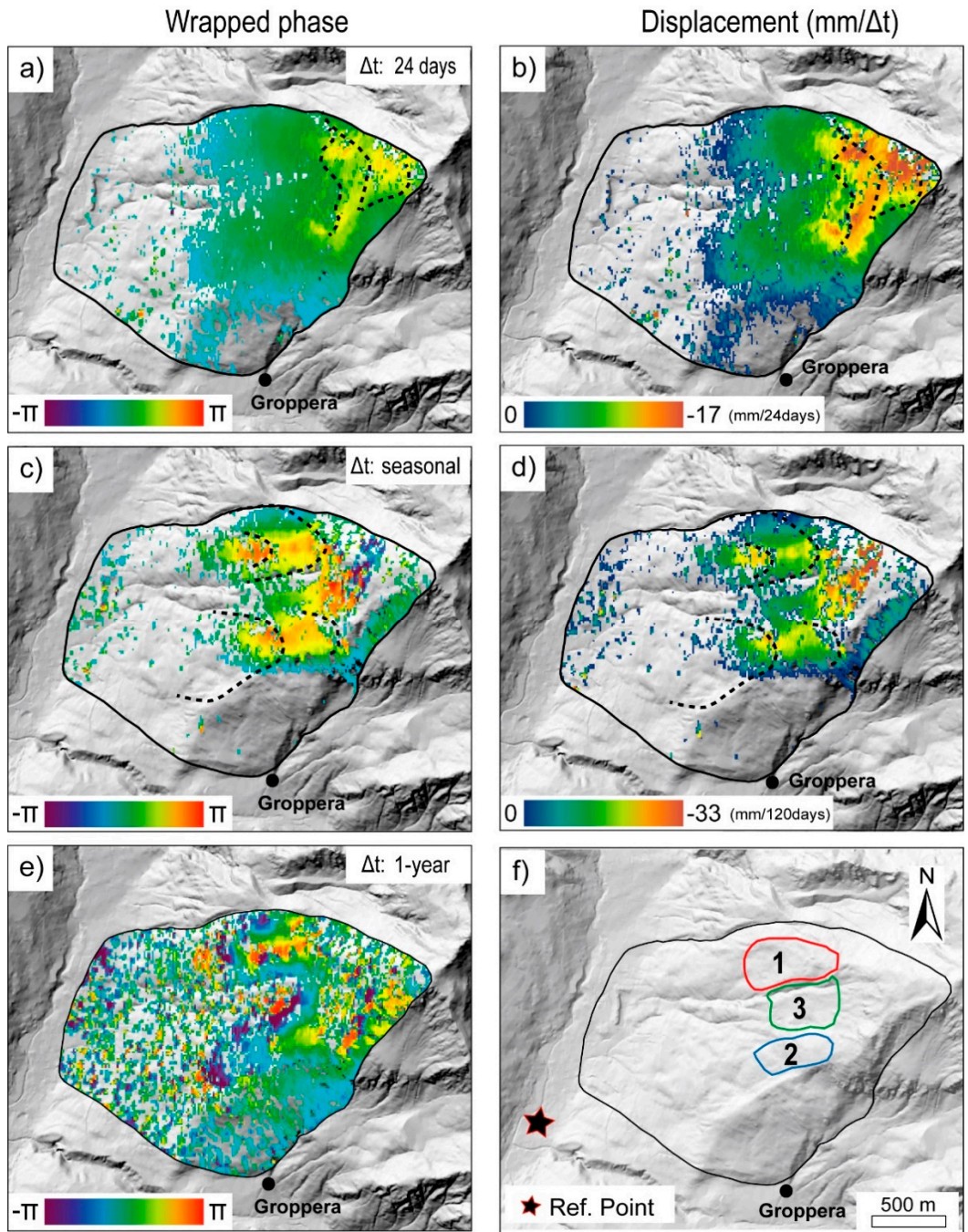

**Figure 4.** Examples of interferograms and displacement maps obtained by DInSAR with different temporal baselines (images: Sentinel-1 (Track 66, descending). Left column: wrapped phase interferograms with increasing temporal baseline: (**a**) 24 days (2–26 July 2017); (**c**) seasonal (June–October 2019); (**e**) 1-year (2018–2019). Right column: (**b**) and (**d**) unwrapped phase converted to displacements (mm). The unwrapped results for the 1-year interferogram were affected by a high noise level and unreliable values, and for reason are omitted; (**f**) sectors 1, 2 and 3, representative of the activity of northern and southern nested landslide heads and the background DSGSD, respectively. 24-days and seasonal wrapped and unwrapped interferograms are masked where the phase coherence of the filtered interferogram is < 0.4 and displacement >0. On 1-year interferograms pixels with coherence <0.3 are masked out. The reference point used for phase to displacement conversion is shown.

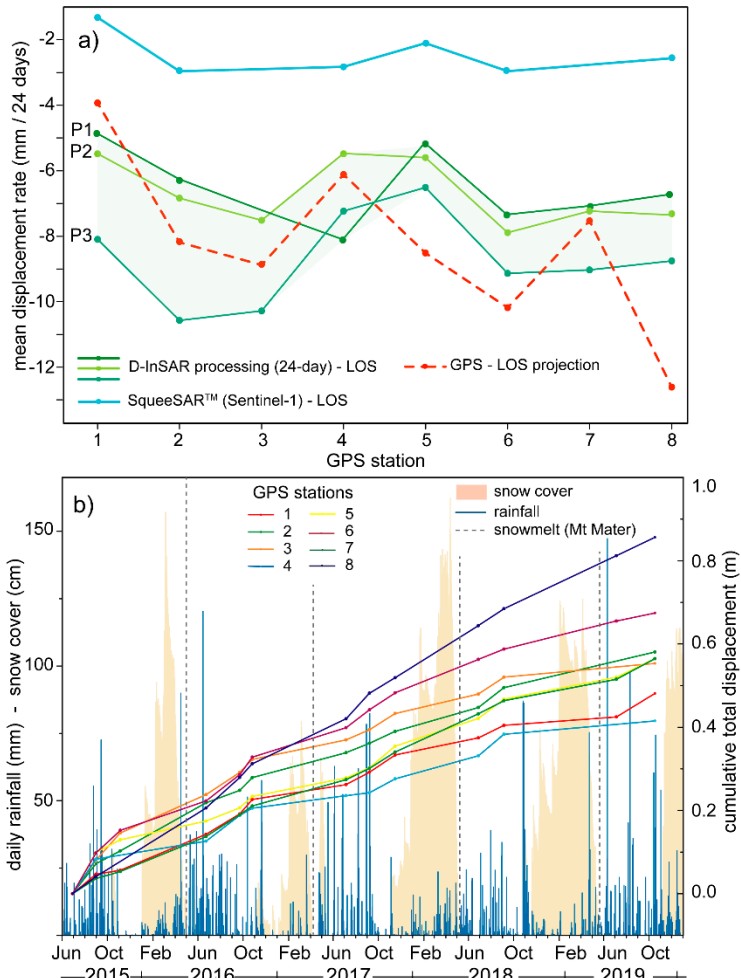

**Figure 5.** Spatial and temporal trends of activity from GPS and DInSAR data. (**a**) spatial trends of displacement rates derived from SqueeSAR$^{TM}$, DInSAR and GPS data, extracted at GPS benchmark locations and scaled to a 24-day period. SqueeSAR$^{TM}$ data were processed over a 2-year acquisition period; GPS displacement rates are derived from cumulative displacements averaged over the 4-year observation period and projected along the Sentinel-1 LOS; DInSAR displacement rates were extracted from three unwrapped 24-day interferograms (P1: 14/06/2017–14/07/2017; P2: 02/07/2017–26/07/2017; P3: 12/09/2017–06/10/2017); (**b**) time series of cumulative GPS 3D displacements compared to daily rainfall and snow cover (data: ARPA Lombardia).

GPS data show evident seasonal trends, characterized by increased displacement rates in the early summer-autumn period (June to October) as an effect of combined snowmelt and rainfall (Figure 5b). In fact, due to the high elevation (2800 m a.s.l.), snowmelt in this sector usually starts around the end of May, later than usually observed at the valley floor. In the snowmelt period, major portions of the slopes are still covered by snow, hampering DInSAR processing [12,37,38] and accurate quantification of snowmelt contributions to displacements.

Seasonal interferograms (Figure 4c) with longer temporal baselines clearly highlight continuous displacement fields, corresponding with: (a) localized deformation along the major DSGSD scarps at 2600 m and 2700 m a.s.l. (Figures 1 and 4d), occurring at rates of several millimeters per seasonal cycle. This signal is very different from that of shallow debris movement detected in 24-day interferograms (Figure 4b) and reflects deep-seated slow deformations; (b) displacements of the head and internal sectors of the northern nested landslide below 2400 m a.s.l. (Figures 1 and 4d), at rates up to 5 mm/month depending on the period (Figure 6); (c) movements up to 3–4 mm/month (Figure 6) of the southern nested landside head at 2100 m a.s.l. and the strongly segmented above sector, up to a

neat curved scarp at 2500 m a.s.l. (Figures 1 and 4d). DInSAR-derived displacement maps report for these sectors velocities ranging from a mean value of 1.5 to 4 mm/month, depending on the considered seasonal period (Figure 6). This suggests a differential seasonal response also for the middle slope sector, with variable trends conditioned by the snowmelt timing and rainfall input in the across winter and springtime.

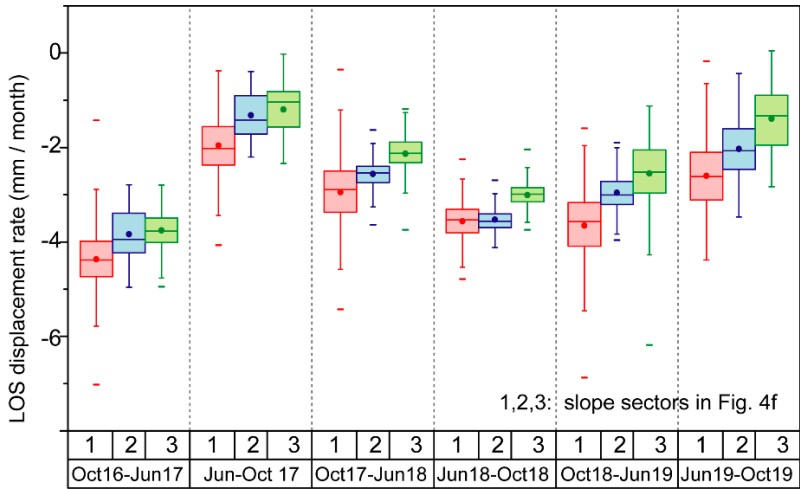

**Figure 6.** Seasonal response of slope sectors. Distributions of LOS displacement rates extracted from seasonal interferograms (June–October and October–June) for the 3 sectors highlighted in Figure 4f. Whisker lengths: upper and lower inner fences based on interquartile range (IQR).

To better constrain this observation, we selected three slope sectors (Figure 4f) corresponding to the nested landslides heads and above scarps (sectors 1 and 2) and the area in between (sector 3). We compared displacement rate distributions between the three sectors during consecutive seasonal periods, from 2016 to 2019. Boxplots in Figure 6 show that displacement rates change seasonally in response to the magnitude and timing of different hydrological inputs, with variable trends through the considered years. In 2016–2017, when snowmelt started early in April (Figure 5b), displacement rates recorded across the snow cover periods (Oct-Jun; about 4 mm/month) were higher than in the summer period (Jun-Oct; about 1.5 mm/month).

An opposite behavior is observed for 2017–2018. In this case, snowmelt occurred later (Figure 5b) and its contribution is recorded by the summer interferogram, with larger displacement rates between June and October (Figure 6). In the 2018–2019 period, despite late snowmelt (May; Figure 5a), abundant rainfall during autumn and spring triggered an anticipated acceleration, with mean LOS displacement rates of about 3.5 mm/month. While consistently reflecting seasonal changes, displacement rates in the three considered sectors confirm a sharp internal segmentation of the DSGSD. The northern nested landslide (sector 1) is always faster than the southern one (sector 2) and both are faster than sector 3, which is interpreted as representative of the background DSGSD activity.

Because of the low signal to noise ratio and poor coherence, we could not derive reliable unwrapped displacement maps from 1-year interferograms. However, wrapped phase maps (Figure 4e) still provide valuable information on the persistent, pluri-annual displacement signal of the DSGSD. In fact, two fringe cycles are evident in the nested landslide sectors and a coherent signal turns out between them in the central part, corresponding to the previously identified sector 3. Such signal unlikely results from shallow movements in the debris cover, that would decorrelate with such a long temporal baseline.

## 5. Discussion

A detailed slope-scale characterization of very slow rock slope deformations like deep-seated gravitational slope deformations [4,29] remains challenging. Successful slope-scale DInSAR

applications have been presented in the literature for slow-moving landslides characterized by displacement rates far exceeding 5 cm/year and relatively homogeneous displacement fields [39–42]. Slower movements typical of DSGSD [13,29] are close to the limits of detection of DInSAR. In fact, DSGSD is characterized by complex mechanisms, reflected by heterogeneous displacement fields and creep at very slow rates. Moreover, these phenomena occur in high mountain environments, where atmospheric and snow-cover disturbances strongly affect the quality of remote sensing products. Although persistent-scatterers interferometry (PSI) techniques proved their ability to measure displacement rates of a few mm/year for coherent targets [12,13] due to their point-like nature, they often fail at capturing a complete picture of the spatial variability associated to slope failure mechanisms.

Our study shows that integrating field morpho-structural observations, remote sensing data from different InSAR techniques and ground-based monitoring allows a successful detail-scale investigation of heterogeneous, very slow-moving rock slope deformations. In particular, we characterized the activity and kinematics of the Mt. Mater DSGSD (Figure 7), wisely using remote sensing techniques to cope with practical difficulties due to: (a) displacement rates close to the limits of remote sensing techniques; (b) strong segmentation of displacement fields, due to secondary large landslides nested at different depths and characterized by different degree of both localized and distributed material damage; (c) lack of subsurface borehole or geophysical investigation data providing constraints to the interpretation of the deep structure of DSGSD.

In the first stage, we tested the potential of available SqueeSAR$^{TM}$ PSI data, that are now used as a standard in the scientific community, for the characterization of very slow rock slope deformations. Performing 2DInSAR analysis to combine information from Sentinel 1A/B ascending and descending orbits, we fully exploited PSI data to retrieve a sound interpretation of the global kinematics of the DSGSD (Figure 3). However, these data have a point-like, sparse nature, making them unsuitable to capture the spatial heterogeneity of the phenomenon. Moreover, SqueeSAR$^{TM}$ data processed over large areas are usually unable to account for the temporal patterns of activity of individual very slow-moving DSGSD and the site-specific atmospheric and snow cover disturbances, which greatly affect the local signal to noise ratio. Thus, we adopted a targeted DInSAR processing approach integrating multi-temporal baseline interferograms (2016–2019) constrained by detailed field morpho-structural observations and GPS data to unravel the deformation patterns and trends of activity in a spatially distributed fashion (Figure 4).

DInSAR processing, specifically designed on the spatial and temporal scales of slope processes associated with DSGSD, provides data consistent with ground-based GPS measurements. As the ability of GPS to measure very small displacements is limited by several error sources such as satellite coverage, temperature changes, and ground dilation, for each GPS benchmark we considered displacement rates averaged over the entire 4-year measurement period. Despite smoothing non-linearities of the GPS time series, this reduces the errors associated with individual measurements and provides a consistent figure of persistent displacement rates. GPS and DInSAR displacement rates consistently settle within about ±2 mm/24 days, according to the specific 24-day period covered by interferograms. Despite keeping the same spatial trend of DInSAR products obtained with targeted temporal baselines, SqueeSAR$^{TM}$ data processed over large areas [37] always underestimate displacement rates (Figure 5a). This might be caused by reduced spatial coherence (quality) of the interferograms included in the SqueeSAR$^{TM}$ analysis (mainly associated with long temporal baselines considered and/or the occurrence of relatively large spatial/ temporal phase gradients), which might result in underestimations of local surface velocities [42].

Selecting image pairs covering different temporal windows (i.e., 24-days, seasonal, annual) allows us to maximize the sensor detection capability for complex phenomena, extracting slope displacement rates, that in the studied case range between more than −15 mm/month and less than −3.5 mm/month. These very different values are typical of associated slope instability processes acting on different scales (e.g., shallow vs deep-seated sliding, nested landslides). Failing to discriminate among these different processes can lead to a misleading interpretation of the overall slope instability mechanisms with major

impacts on the assessment of risk components (e.g., scenario volumes, intensity), that can be strongly over- or underestimated. Moreover, 24-day and seasonal interferograms effectively highlighted active sectors characterized by different spatial patterns and rates of displacements.

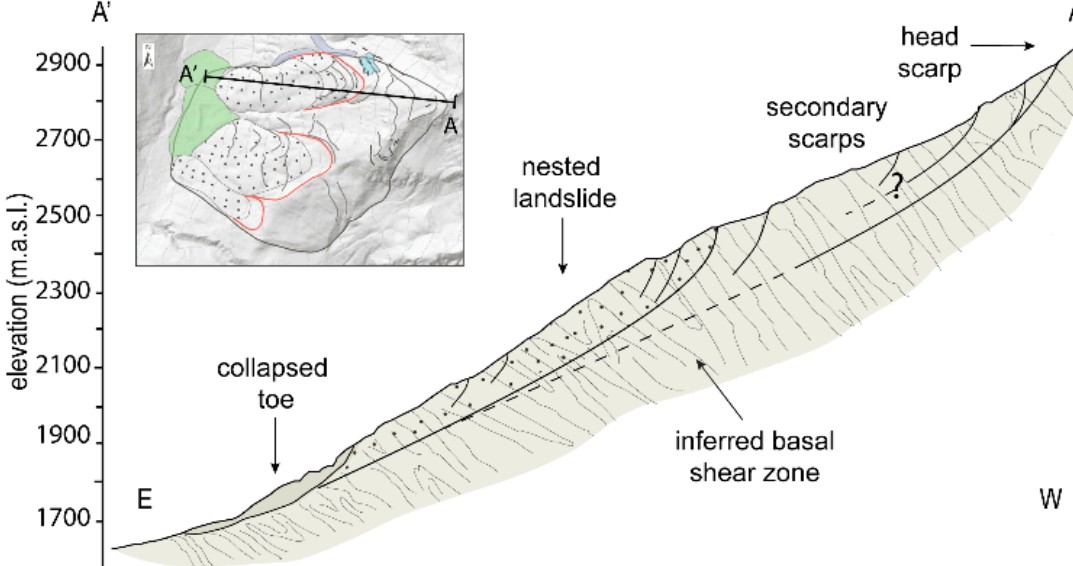

**Figure 7.** Interpretative cross-section across the northern sector of the Mt. Mater slope. DSGSD morpho-structures, basal shear zones and nested landslides are reconstructed from field evidence and interpretation of DInSAR displacement patterns. Cross-section trace is shown in the inset.

Combining mapping and morpho-structural information with remote sensing constraints on kinematic and spatial segmentation of the slope, we were able to provide an interpretative geological model of Mt. Mater slope (cross-section in Figure 7), even without the support of investigations. Different morpho-structures and their associations are witnesses of different deformation mechanisms and provide a first insight into the kinematic behavior of the landslide. We mapped several orders of persistent scarps that dissect the slope from the edge to the toe and only a few minor counterscarps that cut the southern border of the DSGSD towards the Groppera valley. This morpho-structural assemblage suggests a translational sliding mechanism characterized by synthetic structures developing at different depth levels and bounding discrete sectors. Field surveys and aerial photo-interpretation revealed that the slope deformation is sliced by secondary large landslides, nested at different depths within the main DSGSD. These are characterized by rotational movements in the head sectors, where the highest vertical displacements are recorded, and by a variable amount of internal damage. The presence of nested sectors reflects a deep complexity of slope deformation mechanisms, controlled by multiple shear zones almost parallel to the slope profile (Figure 7).

While 1-year interferograms provide a picture of long-term background DSGSD displacement signals, the combined analysis of seasonal interferograms and GPS data outline a sensitivity of the different slope sectors to hydrological forcing. Displacement rates in the middle-upper slope are dominantly sensitive to snowmelt, modulated by spring rainfall depending on the relative timing and magnitude of the two contributions. This induces differential responses across the snow cover period and the summer season. The large landslides nested in the main DSGSD mass exhibit similar styles of activity, but deform at displacement rates higher than the background signal of the DSGSD.

This may be due to the development of different shear zones at shallower depth from the toe to the top and their different degree of evolution that condition the response of the relative slope sectors to hydrological inputs.

Our approach, that can be applied also to other slow rock deformations in different geological and geomorphological settings, proved to supply key information (i.e., internal segmentation, style of

activity, forcing) required to define reference scenarios for risk analysis and mitigation of a widespread, yet challenging class of slope instabilities.

## 6. Conclusions

Very slow rock slope deformations (e.g., DSGSD) in high mountain environments are challenging phenomena for InSAR applications, due to slow movements (usually less than 5 cm/year), spatial and temporal heterogeneity and strong impacts of atmospheric and snow-cover disturbances. We show how these issues can be handled through an integrated, targeted study approach combining field morpho-structural mapping, DInSAR analyses, and GPS data.

In this context, PSI data are very useful for a first-order characterization of slope activity and to retrieve its kinematics, but often fail to capture spatial segmentation, temporal trends and associated mechanisms for site-specific applications because of their point-like nature. DInSAR processing, targeted on multiple temporal baselines, allows us to successfully unravel the mechanisms, temporal trends of activity and forcing factors of heterogeneous DSGSDs characterized by discrete slope sectors possibly undergoing different evolution. Nevertheless, a sound interpretation of these remote sensing results requires detailed geological and morpho-structural field constraints.

Our observations depict a complex phenomenon characterized by nested sliding surfaces bounding discrete sectors that show different sensitivity to hydrological triggers and possibly undergoing different trends towards failure.

**Supplementary Materials:** The following are available online at http://www.mdpi.com/2072-4292/12/8/1329/s1, Table S1: List of images—SqueeSAR ascending dataset, Table S2: List of images—SqueeSAR descending dataset, Table S3: Interferograms derived by Sentinel-1 SLC image processing.

**Author Contributions:** Conceptualization, F.A. and C.C.; Data curation, C.C., F.F. and F.A.; Investigation, M.Z. and F.A.; Validation, G.B.C. and L.D.C.; Writing—original draft, C.C. and F.A.; Writing—review & editing, A.M., F.A., C.C. All authors have read and agreed to the published version of the manuscript.

**Funding:** This research was funded by Fondazione Cariplo, Grant 2016-0757 Slow2Fast to F. Agliardi.

**Acknowledgments:** We thank ARPA Lombardia for providing GPS data and Margherita Spreafico and Elena Valbuzzi for the fruitful discussions.

**Conflicts of Interest:** The authors declare no conflict of interest.

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
