# Peer review of "Unraveling Spatial and Temporal Heterogeneities of Very Slow Rock-Slope Deformations with Targeted DInSAR Analyses"

_remotesensing, doi:10.3390/rs12081329_

Round 1

Reviewer 1 Report

The author did a lot of work in the study of Deep-Seated Gravitational Slope Deformation (DSGSD). They analyzed the slope kinematics by SqueeSAR results and retrieved the spatial displacement pattern by DInSAR results. Then they combined the GPS data with these results to analyze the seasonal displacement pattern. This article is well organized and I am happy to see it published. However, the significant innovations on technology and applications are not clearly, the authors should improve it.

Having said that, this manuscript suffers from numerous English grammatical errors, such as the sentence “The evolution ... , and occur through progressive damage accumulation.” And many long sentences are hard for readers to grasp their meanings. I suggest the author to pay more attention on English editing.

The author used the SqueeSAR results and DInSAR results. I suggest to evaluate the accuracy and explain why the SqueeSAR results are obviously smaller than those results measured by other methods.

It is not my intention to discourage the author that the innovation of this manuscript is not clear. Although it is an integrated study, I prefer to see more innovation in technology or application. Maybe add some work to the selection strategy of the interferograms to achieve an automated pairing?

Detailed Comments

Page 1, line 26: What does “strong internal complexity” refer to?

Page 1, line 30-33: It’s too long. Maybe break into several sentences. May change “at rates one order of magnitude lower than usually considered” to “with velocities of about xx mm/a”.

Page 1, line 40: Change “occur” to “occurs”.

Page 2, line 45-46: May replace “uo to few cm per year” with “smaller than xx cm/a” ? And there are multiple subjects in this sentences. Please rewrite it. Maybe break into several sentences.

Page 2, line 62-63: This sentence is not to the point. I think the main challenge of PSI application in mountain environments is the sparse measurement results. In most cases, they are not dense enough to analyze the spatial displacement pattern. Rewrite this sentence.

Page 2, line 65: Is “targeted DInSAR” a new method the author proposed? If not, please add its reference.

Page 3, Fig1 (a): Add latitude and longitude coordinates.

Page 4, line 128: Change “Materials” to “Data”.

Page 4, line 129: Dose “different data” means “data from different satellites” or “data processed by different methods”?

Page 5, line 144: delete the first “the”. Add the “direction” after the “LOS”.

Page 5, line 145: Change “Measured displacement can ... slightly underestimated” to “Therefore, the measured displacements can be regarded as reliable results although with a slight underestimation (smaller than xx mm/a). ”

Page 5, Fig3: The author said they used SqueeSARTM data from both ascending and descending orbits (see line 137-141). Fig3 only shows the result of descending data. Please add a figure to show the ascending result.

Page 5, line 157: How are these “pseudo-PS” points selected? My guess is that they are points that are observed in both ascending and descending results. Rewrite relevant sentences to be clearer.

Page 5, Equation (1) - (2): As the LOS displacements (Va, Vd) are from different orbital data, their different heading angle provide the potential to retrieve the 2D deformation field. I think the heading angle is an important factor. Why does the author neglect this factor in the calculation? 

Page 7, line 186-188: In Fig4.d, there are several areas which has obvious phase abrupt changes (red and blue handover). Is this result reliable? Please explain the phenomenon.

Page 7, line 202: Rewrite the “providing ... kinematics.”. Its expression is not good.

Page 9, line 242: What does “tim” mean? Does the author means “time span”?

Page 10, Fig5 (a): Why the SqueeSAR results are obviously smaller than those results measured by other methods? Please explain the reason in the manuscript.

Page 11, line 287-292:

The authors said that the result in Fig4.d is not reliable. Is there any other evidence to support this conclusion? Perhaps the coherence map will explain the conclusion better.

Reviewer 2 Report

The manuscript topic fits well with the scope of the remote sensing journal. The authors investigated and analyzed the sentinel data to characterize the spatial distributions of DSGSD using displacement patterns based on SqueeSAR technique and multi-temporal interferograms with various temporal baseline. By controlling the temporal baselines, the authors were able to emphasize and outline the characteristics of the landslides.

As a general comment, I would like to thank the authors for the work. This manuscript has profound geological backgrounds of the study site. I enjoyed a lot. Also, it was very interesting to see the way you linked the interferograms to the geological interpretation.

However, while I read the manuscripts, small doubts raised in my mind (see comments). These may be minor questions and comments, but it may take some times to answer question and modify your manuscript if you agree with my opinions. This is why I chose the “major revision’.

Line 52 : add “ and/or”  strain partitioning, damage localization, and/or secondary landslides….

Line 138: Sentinel 1a-1b à Sentinel 1A/B

Line 144 : the author need to clarify the sentence.

 First, the authors used ascending and descending data for SqueeSAR analysis, but this sentence seems to explain only descending only. The considerations of the ascending should be described as well. Also, the authors may need to make sure why the displacement may be underestimated. The displacement components in 3D domain are projected in LOS vector in SAR interferometry. Thus, the displacement perpendicular to the LOS cannot be measured in LOS axis. However, I think this is because 3D displacement is just projected to LOS not underestimated. So “less sensitive” may be more appropriate rather than “underestimated”.

Figure3. The authors may need to consider to put the SqueeSAR result for ascending orbit in Figure 3. I guess the author thought there is no need to add ascending PS figures because 1) sufficient number of PSs was not detected through the ascending SqueeSAR method and/or 2) ascending PSI is less sensitive to the true displacement in your AOI. But adding the figure may be beneficial to support why you need to apply “targeted DInSAR”.

Section 3.1.

The authors should include more information about the number of scenes for SqueeSAR analysis and whether you excluded the snow-covered scenes or not. In section3.2. the authors explain why the authors have chosen some interferometric pairs rather than all interferometric pairs, but I was not able to see any similar comments for squeeSAR. Please revise this section.

Section 3.2.

Please add why the authors excluded the ascending interferograms in the analysis. In section 3.1, the authors decomposed the velocity maps with two different geometries to 2D vectors (EW and vertical). Similarly, you can try 2D decomposition with targeted DInSAR methods.

Line 183-184: It seems that the authors selected only one pair for the seasonal interferograms. Unlike to time-series InSAR methods (PSI, SqueeSAR, SBAS, etc..), there is no way to mitigate the atmospheric noise from the individual interferograms. Of course, you may try the weather forecasting model such as WRF, or calculate some delays based on weather reanalysis data (ECMWF). If the authors thought that effect of the atmosphere may be negligible, I hope the authors mention the expected amount of the atmospheric phase delay. Usually, since the stratified atmospheric phase delay is a function of the topographic elevation and related to seasonal change, it may not be negligible.  

Please see these papers:

Doin et al. Corrections of stratified tropospheric delays in SAR interferometry: Validation with global atmospheric models

Jung et al. Correction of atmospheric phase screen in time series InSAR using WRF model for monitoring volcanic activities

Line 212-214: This comment is related to comment in section 3.1. The number and density of PSs are the essential factor for comprehensive and reliable analysis. I agree with the authors’ opinion. But, I have small doubts how you produced SqueeSAR result. Throughout the whole paper, I was not able to find some descriptions that the authors excluded the snow-cover interferograms for SqueeSAR. Could you explain more? If the snow-cover scenes were included in the squeeSAR processing, it definitely affected the temporal coherence and quality of the time-series displacement ( and velocity).

In addition, it sounds like you additionally produced the “targeted DInSAR” because SqueeSAR was not successful enough in comparison with GPS data (figure 5a). Based on your analysis and table 1 you listed, I guess that small baseline subset (SBAS) algorithm may be more appropriate rather than PSI or SqueeSAR. You can preserve the spatial coherences using the interfereometric pairs with some small temporal baselines. Would you be able to explain why you designed this analysis?

Line 251 and figure 5. Did you plot the figure 5b with sqrt(ew^2+ns^2+vert^2) of GPS displacement measurements?

Figure 6 and line 278- . You may need to add more explanation here. From figure 6, the “opposite behavior” sounds a reasonable explanation. However, in comparison with figure 5 (GPS measurement), it looks like the displacement rate in June 18-Oct 18 is not smaller than Oct 17- June 18.  Is it because of different locations of GPS measurements and secsion1-3?

Line  309. I have a small doubt whether the authors actually “overcome” the limitation of remote sensing capability. I think the authors “wisely used” the remote sensing technique.

Line 322-324. I don’t agree with this sentence. I believe this is a matter of how you design and process the SqueeSAR or PSI techniques rather than the general property of PSI. You cannot generalize this.

Reviewer 3 Report

The paper describe a detailed study of a Deep-seated gravitational slope deformation DSGSD through DInSAR methods and GPS. Authors seek to unravel the kinematics, segmentation and style of activity of the studied DSGSD. They achieve successfully their objectives combining GPS measurements, point-like InSAR data derived from SqueeSAR method and spatially-distributed information from processed multi-temporal baseline interferograms. In my opinion, the most striking feature of the paper is the use of interferograms to detect very slow movements in the DSGSD, attempting to distinguish between the movement of the entire rock slope failure and several nested landslides. Only for this point I consider that the manuscript must be accepted because demonstrate how to overcome the limitation of InSAR techniques in measuring the movements in a complex landslide such as DSGSDs. However, I recommend the manuscript for publication after moderate revision:
1. The English is good but the text requires restructuring sections 3 and 4.
- The Figure 3 must be in the section 4 because represents the results gathered through the SqueeSAR method.
- I think that you do not need to reference figures of section 4 in section 3.
- I am not sure if the study includes a geological and geomorphological field survey. According to the Authors Contributions section, Mattia Zonca has carried out this task and in line 306 it is stated "Our study shows that integrating field morpho-structural observations, remote sensing data...". Therefore, is the information of section 2.3 results of the study too? If that is the case, authors must describe how they have developed that task and move section 2.3 to section 4. This work is essential for a sound interpretation of the observed kinematics and needs to be enhanced.
2. I consider that it would be interesting to discuss about the observed kinematics and its plausible explanation taking into account the geological and geomorphological information. Authors only showed their results and focused the discussion on the applied techniques. However, some audience would like to know what authors believe about the fastest sector is in the upper part of the slope. How do you explain that?
3. The conclusion section is very vague. Please, rewrite this section in a more specific manner.

OTHER REMARKS:

- Line 52. A citation/s could complete the sentence that end with "to external forcing".
- Figure 1A: Please, include a subfigure of the geographical setting (for example, at Italy-Switzerland scale) to locate the Valle Spluga and Madesimo village for an easier identification of your study area.
- Lines 143-144: Please, correct the sentence “The Mt. Mater the slope is favourably…”.
- Line 210. Space between "30" and "mm/y".
- Figure 7: This is a result and should be mentioned in the Results section.
- Line 354: Mistake. You refer to Figure 7, not 6.

Round 2

Reviewer 1 Report

The revised paper has a great improvement, it can be accepted now.

Reviewer 2 Report

Dear authors, 

Thank you for the answers in detail. I found that this version of the manuscript is improved a lot comparing to the previous version. My question and comments were all resolved. So, I think the manuscript can be published. I appreciate your efforts to make it better. 

Best regards, 

Reviewer.